# Physico-Chemical Properties of Lithium Silicates Related to Their Utilization for Concrete Densifiers

**DOI:** 10.3390/ma16062173

**Published:** 2023-03-08

**Authors:** Lukáš Kalina, Vlastimil Bílek, Martin Sedlačík, Vladislav Cába, Jiří Smilek, Jiří Švec, Eva Bartoníčková, Pavel Rovnaník, Josef Fládr

**Affiliations:** 1Faculty of Chemistry, Brno University of Technology, 612 00 Brno, Czech Republic; 2Faculty of Civil Engineering, Brno University of Technology, 612 00 Brno, Czech Republic; 3Faculty of Civil Engineering, Czech Technical University in Prague, 166 36 Prague, Czech Republic

**Keywords:** concrete densifier, lithium silicate, surface treatment, gelation process

## Abstract

Protection of concrete against aggressive influences from the surrounding environment becomes an important step to increase its durability. Today, alkali silicate solutions are advantageously used as pore-blocking treatments that increase the hardness and impermeability of the concrete’s surface layer. Among these chemical substances, known as concrete densifiers, lithium silicate solutions are growing in popularity. In the present study, the chemical composition of the lithium silicate densifiers is put into context with the properties of the newly created insoluble inorganic gel responsible for the micro-filling effect. Fourier-transform infrared spectroscopy was used as a key method to describe the structure of the formed gel. In this context, the gelation process was studied through the evolution of viscoelastic properties over time using oscillatory measurements. It was found that the gelation process is fundamentally controlled by the molar ratio of SiO_2_ and Li_2_O in the densifier. The low SiO_2_ to Li_2_O ratio promotes the gelling process, resulting in a rapidly formed gel structure that affects macro characteristics, such as water permeability, directly related to the durability of treated concretes.

## 1. Introduction

Alkali silicate solutions are widely used in various applications. One of them is their utilization in the building industry as concrete densifiers. They can effectively block the capillaries existing in the concrete surface, resulting in an increase in the hardness and impermeability of the concrete’s surface layer [1]. In addition to the traditional use of sodium and potassium silicate solutions, densifiers based on lithium silicates have been growing in market share. Among the main advantages of lithium densifiers are lower alkalinity [2], higher water resistance, prevention of alkali-silica reaction [3,4], as well as a potentially lower tendency to efflorescence compared to sodium and potassium alternatives [5].

The widely accepted working mechanism of alkali silicate densifiers lies in their reaction with calcium ions in the surface layer of the concrete to form an inorganic gel [6]. The formed insoluble silicate gel acts as a micro-filler, generating a compact and dense microstructure on the treated concrete surface. Subsequently, this affects many characteristics of concrete, such as a decrease in water absorption [7,8], chloride permeability [6,9], and carbonation depth [6,10]. Simultaneously, a noticeable increase in abrasion [11,12] and frost resistance [6] has been reported.

In addition to the quality and properties of the concrete substrate or ambient conditions, the chemical composition of the used alkali silicate is of great importance. The main factors are primarily the cation nature, the total SiO_2_ content, and the silicate modulus (molar ratio between SiO_2_ and alkali oxide). These main chemical parameters play an important role in gel formation induced by the addition of calcium ions. The effect of the ion type is fundamentally connected to their ion radius. The increase in cation size results in a more efficient structural disorder, which is then responsible for a long gelation process [13]. Moreover, according to the hydration rule, the small cation binds water more strongly and has a smaller effective charge than a large cation with the same valence. Thus, large potassium ions are more effective in compensating the charge of the silicate anions than sodium or lithium ions, which cause gelation to be slower [14]. Hence, in the case of lithium silicates, polycondensation reactions are more preferred. The effect of the total SiO_2_ content is subject to the following trend. The time of gelation increases with decreasing silicon concentration in the solution due to the dilution of the reactional medium. In such a system, the silanol tetrahedra units are strongly dispersed and the polycondensation reactions become difficult, inducing a long gelation time [15]. On the other hand, a silicate concentration that is too high is responsible for the increase in viscosity of the used densifier, which affects the ability of the molecules of alkali silicate to make adequate contact with calcium ions, causing a prolongation of gelation times [8]. Finally, the effect of the silicate modulus can significantly influence the point of gelation. The concentration of alkali ions has a direct impact on the electrostatic repulsion forces among the silica units. When the concentration of cations increases (reducing the silicate modulus), the zeta potential value of the originally negatively charged silica species becomes less negative and approaches the isoelectric point, which is related to the aggregation and acceleration of gelling [16].

The time of gelation determines the correct function of the used densifier. As previously stated, a gelation time that is too short results in a low penetration of the densifier into the sample, because its pores are blocked quite fast [8], while a gelation time that is too long is not desirable from a technological point of view. Previous studies have mainly solved the effect of the cation nature [13,17], pH [15], temperature [18], or the amount of dissolved silicate species [17] on the gelation mechanism. In addition, they have not focused on relating the gelation time to the densifying properties of a lithium silicate densifier. For these reasons, this study investigates the influence of the silicate modulus of the used lithium silicate on the gelation process, while keeping the amount of dissolved SiO_2_ the same. Subsequently, the physico-chemical properties of formed gels were put into context with the quality of the surface treatment, which was verified by means of water permeability tests.

## 2. Materials and Methods

### 2.1. Preparation of the Lithium Silicate Densifiers

Lithium silicate solutions (densifiers) with different silicate moduli (M_S_ = 3.0; 3.2; 3.4; 3.6; 3.8; 4.0) were prepared by mixing commercially available lithium water glass (M_S_ = 4.51) with the appropriate amount of lithium hydroxide monohydrate to obtain the corresponding modules. An example of the preparation of lithium silicate densifiers (LiSD) is presented in Table 1. The total SiO_2_ content in all lithium densifiers was held at 18 wt%. The chemical composition of lithium waterglass (SChem, a.s., Ústí nad Labem, Czech Republic) as well as lithium hydroxide monohydrate of analytical grade purity (Penta, s.r.o., Prague, Czech Republic) was determined using conductometry titration and is included in Table 1.

### 2.2. Preparation and Surface Treatment of Cement Mortar Samples

Ordinary Portland cement (OPC CEM I 42.5 R) mortar samples with dimensions of 4 × 4 × 16 cm^3^ were prepared for their subsequent surface treatment using lithium silicate densifiers. The preparation process was inspired by the EN 196-1 standard [19]. The sand-to-OPC ratio was 3:1 using three different fractions of standard siliceous sand, and the water-to-OPC ratio was set at 0.50. A laboratory mixer was used for the mixing. The prepared mixtures were cast into steel molds and moistly (RH~99%) cured at laboratory temperature (25 °C) for 24 h. After the demolding process, the specimens were stored in water at 25 °C for 28 days.

After 28 days, the mortar samples were left in laboratory conditions for 24 h. Subsequently, they were immersed in the lithium silicate densifier for 24 h, ensuring penetration to its maximum amount. At the same time, the reference samples were not put in a densifier, but into water. Thereafter, the samples were removed from the treatment bath and the surface was freed of excess densifiers. After 24 h, the treated and untreated (reference) samples were subjected to a water permeability test.

### 2.3. Chemical Characterization

#### 2.3.1. FT-IR Measurement

Fourier transform infrared spectroscopy (FT-IR) spectra were obtained using a Nicolet iS10 FTIR spectrometer (Thermo Fisher Scientific, Middleton, WI, USA) using the attenuated total reflection method (ATR). Spectra were recorded in the 400–4000 cm^−1^ range with a resolution of 2 cm^−1^. Five separate measurements were carried out for each sample, where every obtained spectrum was the average of 64 scans. The analyses were carried out in an air atmosphere. The background spectrum was measured using the same parameters and was subtracted from the sample spectra. FT-IR was chosen as a suitable analytical technique for the chemical characterization of lithium silicate densifiers. Moreover, the chemical nature of silanol tetrahedra units in lithium silicate solutions was revealed using NMR spectroscopy, which had a significant impact on the evaluation of FT-IR spectra.

In addition, FT-IR was also used to monitor the evolution of the gelation process as a result of the presence of calcium ions. These gels were prepared based on JC/T 1018-2006 standard, where 0.593 g of Ca(OH)_2_ was mixed with 18 mL of deionized water, and then 25 mL of lithium silicate densifiers (the samples varied in silicate moduli–M_S_ = 3.0; 3.2; 3.4; 3.6; 3.8; 4.0) were added to the solution. This mixture was homogenized by manually shaking for 1 min. In this way, the gelation process was initiated and the samples were measured first, shortly before the gelation point, determined according to rheological investigations, and then the gels were measured after 24, 48, 96, and 168 h.

#### 2.3.2. NMR Measurement

NMR spectra were measured on a Bruker Avance III HD 700 MHz spectrometer operating at 139.2 MHz for ^29^Si using a 5 mm dual broad-band probe. To improve the base line and allow signal quantification, the ring-down elimination pulse sequence was used as described by Schraml et al. [20]. Prior to the reading pulse, a 500 μs adiabatic inversion Chirp pulse over 60 kHz was applied. To eliminate the signal originating from the glass in the NMR tube and probe parts, the FID measured with a sample containing 100 mL solution of KCl in D_2_O was subtracted from the data after the Fourier transform. All spectra were recorded at 298.2 K. The spectra were measured using a spectral width of 200 ppm (27.8 kHz); WALTZ16 proton decoupling was applied during acquisition, but not during the relaxation delay. The relaxation delay was set to 10 s and the number of acquired points was set to 64 k. Altogether, 2048 or 4096 scans were collected, resulting in measuring times six and a half or thirteen hours per spectrum, respectively. A mild line broadening of 3 Hz was used in data processing.

### 2.4. Rheological Investigation

Rheological measurements were performed using a rotational rheometer DHR-2 (TA Instruments, New Castle, DE, USA). Samples were prepared as outlined in Section 2.3.1. All measurements, including sample preparation, were carried out at a temperature of 25 °C. The gel point was determined using vane-in-cup geometry. The vane had a diameter of 20.0 mm and a height of 19.5 mm, while the cup diameter was 30.4 mm. The operating gap was 30.0 mm. The measurement was carried out within the range of small-amplitude oscillatory shear (SAOS), i.e., within the linear viscoelastic region (LVR), using a time sweep measurement. The strain amplitude and frequency were kept constant at 0.1% and 10 rad/s, respectively; five cycles per each point. The gel point was evaluated as the crossover of the storage modulus (G′) and the loss modulus (G″), i.e., the point of change in the viscoelastic character of the sample from liquid-like (G′ < G″) to solid-like (G′ > G″). In this case, only one sample of each composition was tested due to high time requirements for measurements.

To avoid the ineffective continuous occupation of the rheometer by repeating the whole gelation process inside the rheometer, the gels were externally prepared in a plastic vial of approximately the same shape as the rheometer cup. The viscoelastic properties of the externally prepared gel samples were determined using amplitude strain sweep measurements in the chosen time intervals (24 and 48 h from the start of gelation). A plate-like sample of each gel was cut off with a scalpel and immediately inserted into the rheometer. A cross-hatched steel parallel plate geometry system 20 mm in diameter was used as a suitable sensor for these types of samples. The advantage of cross-hatched geometry lies in the prevention of wall slip during rheological tests. The frequency of oscillation was maintained at a constant value (1 Hz), while the amplitude of deformation was logarithmically increased from 0.01% until 1000% (logarithmic sweep, 20 points per decade). Before each individual measurement, a conditioning step was included (2 min of sample relaxation at constant temperature 25 °C). The axial force during the squeezing of the gel sample to the geometry gap did not exceed 8 N. The measuring gap was maintained at a constant value (1000 μm). The crucial viscoelastic parameters (elastic moduli in strain independent part, the end of linear viscoelastic region) were calculated as the average (with standard deviation) of at least two to four individual measurements.

### 2.5. Water Permeability Measurement

The water permeability test was performed on the basis of the EN 1062-3 standard [21]. The treated and reference samples were weighed to the nearest 0.1 g. The specimens were inserted into a plastic rack, which ensured maximum contact with the test medium, and immersed in the container with deionized water. The specimens were removed from the container in time intervals of 10 min, 30 min, 1 h, 2 h, 3 h, and 24 h. They were always carefully wiped, weighed, and returned back to the water bath. Water permeability was determined by the change in mass and compared in percentage terms to the reference samples

## 3. Results and Discussion

### 3.1. Characterization of Lithium Silicate Densifiers

The lithium silicate densifiers prepared according to Table 1 were first characterized by FT-IR in ATR mode to identify different vibration bands. Figure 1a shows their FT-IR spectra having a similar nature. The broad band located from 2500 to 3700 cm^−1^ corresponds to both the contribution of OH vibrations in the Si–OH bonds and in water molecules [22]. For water, an intense band at 1638 cm^−1^ as a result of O–H–O scissors-bending is also typical [23]. The asymmetric stretching of the Si–O–Si bonds falls within the area of the extended band from 950 to 1250 cm^−1^ [17]. This broad band with a sharp peak at 1012–1020 cm^−1^ has its origin in the presence of Q^2^ silicate tetrahedra units [24]. The peak is shifted to a lower wavenumber with decreasing silicate modulus relating to the bond strength between the alkali cation and non-bridging oxygen (Si–O^−^M^+^) [13,17]. This statement was also confirmed by the NMR results (Figure 1b, Table 2), where the portion of Q^2^ units representing Si–O–Si and/or Si–O^−^M^+^ connections decreased with increasing silicate modulus at the expense of Q^3^ and Q^4^ cross-linking silica tetrahedra. In addition, the asymmetric stretching broad band is characterized by a distinctive tail at high wavenumbers (1060–1250 cm^−1^), exactly identifying the presence of Q^3^ and Q^4^ tetrahedra [24,25].

### 3.2. Monitoring of Gelation Process by FT-IR

During the gelation process initiated by the presence of Ca^2+^ ions (prepared according to the JC/T 1018-2006 standard), polycondensation reactions occur, and thus the final structure of the silicate gels is formed. These gels are created by the similar chemical mechanism that is given by the cementitious environment, and therefore this approach can simulate the situation inside the treated samples. The main goal of this procedure was to reveal the changes in the chemical composition of gels prepared by the use of lithium silicate densifiers with various Li_2_O contents (silicate modulus).

The FT-IR results of all studied lithium silicate gels are very similar to those of lithium silicate densifiers, including the contributions of water and Si–OH bonds in higher wavenumbers. However, they differ in the region of low wavenumbers, as shown in Figure 2a. The appearance of three new bands is observed. The less noticeable band (**1**) at 577 cm^−1^ is attributed to the O–Si–O bending vibrations. Several studies have reported [13,24,26] that this band is related to the ring structure of silica units, as well as the second more visible band (**2**) of Si–O–Si symmetric stretching vibration at 771 cm^−1^. Finally, the band (**3**), forming a shoulder at 869 cm^−1^, is assigned to the stretching vibration of Si–O^−^M^+^ contributions. While in the case of lithium silicate solutions this type of bonding only affects the shift of the sharp peak associated with Q^2^ silicate units, a noticeable band is observed in the gel spectra. This may be related to the fact that due to the increase in OH^−^ concentration in aqueous solution caused by the addition of Ca(OH)_2_, the hydrolysis of Si–O–Si bonds takes place [27]. As a result, the Si–O^−^M^+^ contributions increase with the appropriate band in the FT-IR spectra, and, at the same time, the wavenumber of the peak representing the Q^2^ entities is strongly affected. Whereas the previous FT-IR measurements of lithium silicate solutions ranged this sharp peak in a broad band from 1012 to 1020 cm^−1^, in the case of lithium silicate gels, it is situated in the area of 963–990 cm^−1^ depending on the different modules. The shift of this band to lower wavenumbers is therefore a consequence of the alkaline cations incorporation (Ca^2+^, Li^+^) into the gel structure, inducing a weakening of the covalent bond for the benefit of the ionic bond, as was previously reported [28,29]. However, during the gelation process, polycondensation reactions take place that increase the number of Si–O–Si bonds and the shift of this band gradually returns back to higher wavenumbers, as shown in Figure 2b. The asymmetric stretching of the Si–O–Si bonds due to Q^2^ and Q^3^ silicate units is located in areas with higher wavenumbers of this broad band, as was previously mentioned for the solutions of lithium silicate densifiers.

### 3.3. Monitoring of Gelation Process by Rheological Approach

The gelation process of the investigated lithium silicate densifiers was monitored using continuous SAOS measurements over time (Figure 3), and discontinuously using strain amplitude sweeps at the selected times (24 and 48 h), from which the storage modulus in the LVR was evaluated (Figure 4). The former approach is advantageous in terms of the continuous monitoring of the change from liquid-like to solid-like behavior, and further subsequent stiffening of the gel. All without the additional need for care regarding the sample and without altering its structure during preparation for the rheometer. On the other hand, continuous measurement is highly impractical in terms of long-term device occupation and unavoidable partial drying, even though the solvent trap is used. Therefore, the combination of these two rheological approaches provides valuable information from the perspective of the gelation process, but also the possibilities of the gel characterization. Figure 3a clearly shows that the gradual increase in the silicate modulus of the lithium densifier from 3.0 to 4.0 greatly prolongs the gel time and also affects the overall evolution of the gel stiffness. As further shown in Figure 3b, the dependence of the gel time on the silicate modulus can be fitted by the simple exponential curve. The relatively low dependence of gel time in the range of relatively low silicate moduli is in agreement with the study by Gaboriaud et al. [30]. In addition, Figure 3a shows similar values of the viscoelastic moduli during their crossover point, regardless of the silicate modulus of the tested densifier. This is likely due to the same mechanism of the gelation process, as already illustrated in Section 3.2. Because the only difference in the compositions of the used densifiers is the content of Li^+^, changes in the dependence of the gel time are likely related to the differences in the ionic strength, pH, and overall capability of Li^+^ to stabilize the silicates. It is worth noting that the long-term stability of the densifiers also increases with increasing silicate modulus.

Figure 4 shows that the mechanical properties of the gels increase over time beyond the timescale of the measurement in Figure 3a. In addition, a clear trend of decreasing the storage modulus with an increase in the silicate modulus of the densifier was verified after both 24 and 48 h. The absolute values of the storage modulus after 24 h seem slightly different for both rheological approaches, which could be mainly related to the sample preparation of the gels for the discontinuous measurements. Nevertheless, these results are well-reproducible, which indicates the suitability of this approach for the long-term characterization of the gels. Finally, it should be noted that both the gel time and the evolution of the mechanical properties correlate well with the trend in the FT-IR results (Section 3.2).

### 3.4. Water Permeability

Since water is essential for many forms of concrete deterioration, resistance to water penetration is a fundamental criterion by which to determine the efficiency of surface treatment [31]. The results of the water penetration test of the treated samples are shown in Figure 5. It can be seen that the rate of reduction in water absorption is highest for samples treated with lithium densifiers, having a low silicate modulus. This fact may seem to go against general theories on how silicates act to improve the performance of concrete. The most often accepted theory on the formation of insoluble calcium-silicate hydrates was mentioned in the introduction. Another theory is based on the precipitation of SiO_2_ from alkali silicate in the pores, and a third standpoint is that the silicates form an expansive gel to fill the concrete voids by swelling [9,12,32]. The important factor in all of them is the presence of SiO_2_ units in the treating medium, and therefore, with their higher amount, its effectiveness should increase, as reported by Jiang et al. [8]. However, this study is focused on the effect of the alkali ion. Thanks to the fact that the SiO_2_ portion is the same for all lithium densifiers (18 wt%), the key parameter is the Li_2_O content controlling the silicate modulus. As already shown in Section 3.3, the Li_2_O amount determines the point of gelation, which is delayed with the use of a higher silicate modulus. One can say that the gelation process, associated with the formation of the gel structure, affects its compactness. Quickly formed structures composed of an amorphous network of silica species significantly slow down the travel of the water molecules passing through the gel, resulting in the water permeability reduction mainly in the initial stages of the test. After a long immersion time (more than 2 h), the molecules of water gradually penetrate into the material and the differences among the samples begin to be marginal. It is important to point out that all lithium silicate densifiers show about 20% better results in comparison with untreated samples after the 24-h water permeability test. This demonstrates the beneficial use of lithium silicate densifiers in the process of concrete surface treatment.

## 4. Conclusions

In this paper, the effect of the silicate modulus of the lithium silicate densifier on its gelation process was investigated. The obtained results were further correlated with the water permeability of the hardened cementitious mortars treated with the densifiers. Based on the obtained results, the following conclusions can be drawn.

For the constant content of SiO_2_ (18 wt%), the gelation time of the lithium silicate densifiers exponentially increased as the molar SiO_2_ to Li_2_O ratio decreased from 4.0 to 3.0. It follows that the lithium silicate solution with a silicate modulus of 4.0 shows a delay of up to 14.5 times in gelation time compared to a solution with M_S_ = 3.0.The storage modulus of the obtained gels also considerably increased with the decrease in the silicate modulus. This trend was observed for both continuous rheological measurements of the gelation process in situ and the measurements of the samples taken from externally prepared gel after 24 and 48 h.The gelation process was accompanied by the shift in the FT-IR band related to Q^2^ units from the wavenumbers of 1010–1020 cm^−1^ to the wavenumbers of 960–980 cm^−1^. The magnitude of this change increased with the decrease in silicate modulus, which correlates with the viscoelastic properties of the gels.The surface treatment of the hardened cementitious mortars with the investigated densifiers led to a decrease in water permeability by 20% after 24 h of the test duration. During the first 60 min, the amount of absorbed water considerably decreased with decreasing silicate modulus, and thus with the gelation time of the used densifiers.

These results emphasize that gelation time is a key aspect of the use of lithium silicates as densifiers for OPC-based materials. To verify it, further research will be focused on the in-depth assessment of the action of the densifiers in a concrete surface, along with their impact on durability.

## Figures and Tables

**Figure 1 materials-16-02173-f001:**
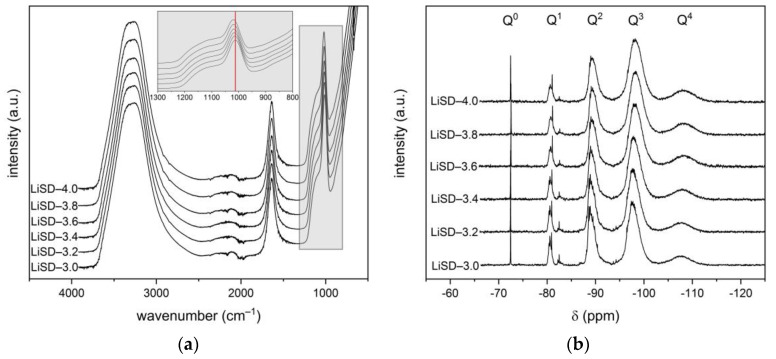
(**a**) FT-IR spectra; (**b**) NMR spectra of studied lithium silicate densifiers.

**Figure 2 materials-16-02173-f002:**
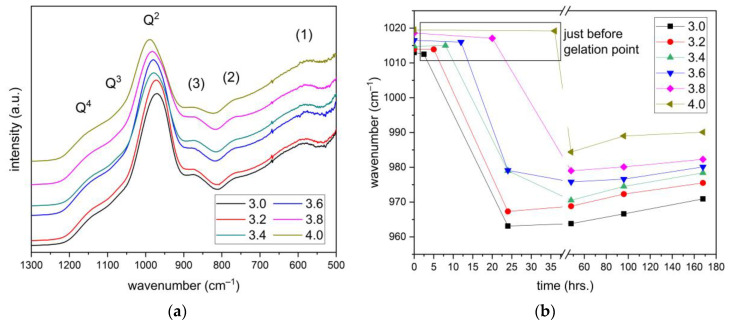
(**a**) Asymmetric stretching vibration bands of silicate entities in lithium silicate gels at the age of 48 h; (**b**) The shift of asymmetric vibration band belonging to Q^2^ silicate units as a function of time.

**Figure 3 materials-16-02173-f003:**
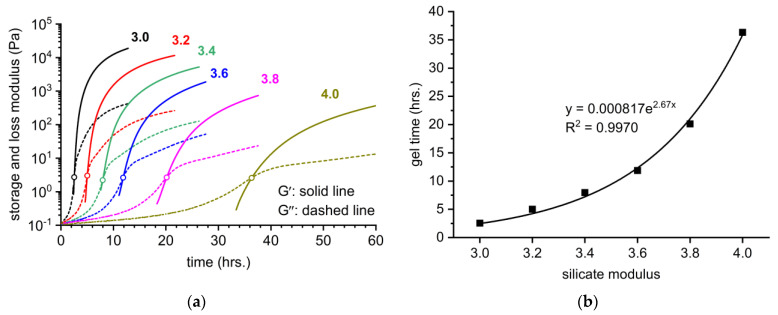
(**a**) Continuous monitoring of the gelation process of lithium silicate densifiers with different silicate moduli. The open circles highlight the gel point, in which G′ = G″; (**b**) Exponential fit of the time needed to reach the gel point plotted against the silicate modulus of densifiers.

**Figure 4 materials-16-02173-f004:**
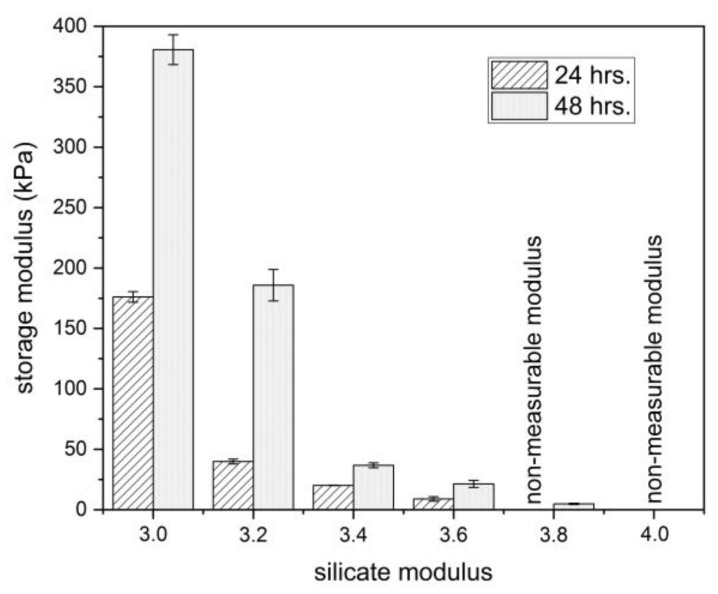
Values of the storage modulus of lithium silicate densifiers with different silicate moduli determined using discontinuous rheological experiments after 24 and 48 h.

**Figure 5 materials-16-02173-f005:**
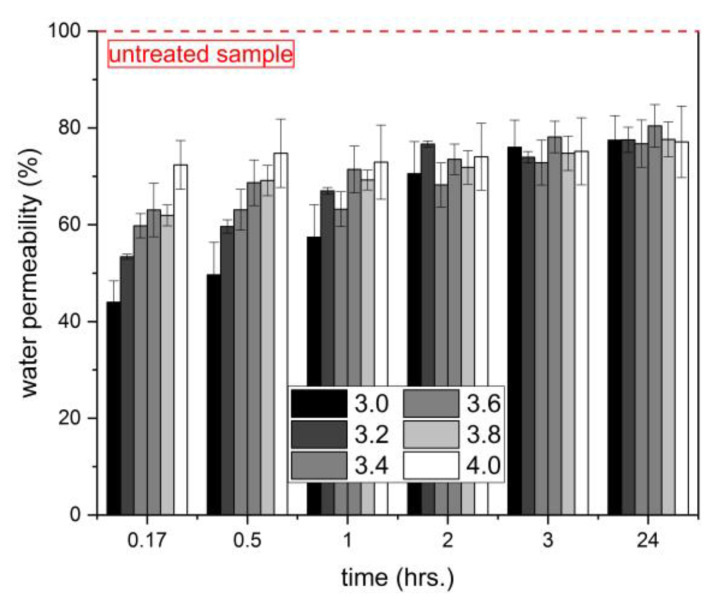
Water absorption of treated samples after applying the lithium silicate densifiers.

**Table 1 materials-16-02173-t001:** Preparation proposal of 100 mL lithium silicate densifiers.

	Quantity (g)		
	Commercial Lithium Waterglass (Li_2_O = 2.1 wt%; SiO_2_ = 19.04 wt%)	LiOH·H_2_O (Li_2_O = 35.61 wt%)	H_2_O
LiSD–3.0	111.55	3.31	3.14
LiSD–3.2	111.55	2.69	3.75
LiSD–3.4	111.55	2.15	4.30
LiSD–3.6	111.55	1.66	4.78
LiSD–3.8	111.55	1.23	5.22
LiSD–4.0	111.55	0.84	5.61

**Table 2 materials-16-02173-t002:** Distribution of Q^n^ silicate units determined by ^29^Si-NMR spectra in lithium silicate densifiers.

Sample/(mol%)	Q^0^	Q^1^	Q^2^	Q^3^	Q^4^
LiSD–3.0	0.90	6.10	28.44	52.74	11.81
LiSD–3.2	0.87	5.77	27.96	53.84	11.56
LiSD–3.4	0.83	5.45	26.37	54.70	12.66
LiSD–3.6	0.83	4.90	25.13	54.98	14.16
LiSD–3.8	0.82	4.70	24.72	55.58	14.17
LiSD–4.0	0.80	4.53	24.23	55.33	15.10

## Data Availability

The data presented in this study are available on request from the corresponding author. The data are not publicly available due to ongoing research.

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
