# Peer review of "Physico-Chemical Properties of Lithium Silicates Related to Their Utilization for Concrete Densifiers"

_materials, 2023, doi:10.3390/ma16062173_

Round 1
Reviewer 1 Report
This manuscript analyzed the chemical composition, gelation process of lithium silicate densifiers and the influence of lithium silicate densifiers on water permeability test of mortar specimen were tested. The research is meaningful to understanding the micro-structure of lithium silicate densifiers. However, because of some shortcomings,I consider this paper to be suitable for publishing after the revisions.
1. The results in abstract and conclusions section are generally introduced. Whether the author could give some more detailed conclusions in these two sections?
2. There are some grammar and other language mistakes in this paper. Please check the mistakes in this paper and then correct it.
3. In “3.1 Characterization of lithium silicate densifiers” section, it lacks unit in table 2.
4. In “2.3.2. NMR measurement “ section, NMR spectra were measured on Bruker Avance III HD spectrometer operating at 700 124 MHz for 1H . There was Nno corresponding results to NMR spectra for 1H in “3. Results and Discussion”section. Please explain.
5.The format of references is not uniform.
Reviewer 2 Report
This paper presents the physicochemical properties of lithium silicates related to their utilization as concrete densifiers. The research is relevant and somehow well organized and the research approach is sound and consistent. The authors reveal a deep understanding of the analyzed issue and have carried out intensive work, and the obtained results might be useful; hence, the paper could be considered for publication. However, the paper cannot be published in its present form because the explanations are extraordinarily obscure. The authors are suggested to resubmit their work after addressing completely the following observations.
1. Some key parameters are not mentioned. The rationale for the choice of the particular set of parameters should be explained in more detail.
2. Please add a sentence or two to clearly recap how your study differs from what has already been done in the literature to ascertain the contributions more strongly.
3. Authors are suggested to mention the test set-up for Chemical characterization and Rheological investigation.
4. Authors are suggested to add a table explaining the number of tests with their details. How many repetitions per test?
5. The manuscript needs to be clarified along with potential implications for
the results on the basis of discussion.
6. Would authors mind explaining the reason for the selection of a specific time interval to determine viscoelastic properties as mentioned on page number 4, lines 149-151.
7. The authors have performed a water permeability test on the basis of EN 1062-3 standard. It should be cited in the text or in the reference list. Similarly, for EN 196-1.
8. The manuscript does not address any identified gaps in the existing literature that this work attempts to fill. The significance of the results is not discussed, and no design considerations are suggested.
9. It is recommended to add microstructure analysis to understand the micro-filling effect and gel formation.
10. The conclusions are not considered totally correct. Moreover, the provided conclusions lack deep conceptual interpretation and this is a major issue and must be fully addressed prior to further consideration.
